# CoADNet: Collaborative Aggregation-and-Distribution Networks for Co-Salient Object Detection

**Qijian Zhang**[1]*, **Runmin Cong**[2,3,1]*†, **Junhui Hou**[1], **Chongyi Li**[4], **Yao Zhao**[2,3]
[1]Department of Computer Science, City University of Hong Kong, Hong Kong SAR, China
[2]Institute of Information Science, Beijing Jiaotong University, China
[3]Beijing Key Laboratory of Advanced Information Science and Network Technology, China
[4]School of Computer Science and Engineering, Nanyang Technological University, Singapore
qijizhang3-c@my.cityu.edu.hk, rmcong@bjtu.edu.cn, jh.hou@cityu.edu.hk,
lichongyi25@gmail.com, yzhao@bjtu.edu.cn
https://rmcong.github.io/proj_CoADNet.html

## Abstract

Co-Salient Object Detection (CoSOD) aims at discovering salient objects that repeatedly appear in a given query group containing two or more relevant images. One challenging issue is how to effectively capture co-saliency cues by modeling and exploiting inter-image relationships. In this paper, we present an end-to-end collaborative aggregation-and-distribution network (CoADNet) to capture both salient and repetitive visual patterns from multiple images. First, we integrate saliency priors into the backbone features to suppress the redundant background information through an online intra-saliency guidance structure. After that, we design a two-stage aggregate-and-distribute architecture to explore group-wise semantic interactions and produce the co-saliency features. In the first stage, we propose a group-attentional semantic aggregation module that models inter-image relationships to generate the group-wise semantic representations. In the second stage, we propose a gated group distribution module that adaptively distributes the learned group semantics to different individuals in a dynamic gating mechanism. Finally, we develop a group consistency preserving decoder tailored for the CoSOD task, which maintains group constraints during feature decoding to predict more consistent full-resolution co-saliency maps. The proposed CoADNet is evaluated on four prevailing CoSOD benchmark datasets, which demonstrates the remarkable performance improvement over ten state-of-the-art competitors.

## 1 Introduction

Different from salient object detection (SOD) that extracts the most attractive regions/objects from a single image, co-salient object detection (CoSOD) aims to discover the salient and repetitive objects from an image group containing two or more relevant images, which is consistent with the collaborative processing mechanism of human visual systems [6, 14, 52]. Mining co-salient patterns across image groups has been widely applied in various down-streaming computer vision tasks, such as image co-segmentation [17, 18, 22], object detection [19, 40], and image retrieval [50]. Conceptually, the co-salient objects belong to the same semantic category, but their category attributes are unknown under the context of CoSOD, which is different from SOD for single image or video sequences [7, 11, 12, 32, 33, 34]. Therefore, co-saliency models are supposed to capture

discriminative inter-image semantic relationships without the supervision of specific category labels, which makes CoSOD a challenging problem especially when facing co-salient objects with distinct appearance variations. Starting from the pair-wise CoSOD model [35], researchers have designed different unsupervised approaches for inter-image relationship extraction, such as clustering [16], rank constraint [3], similarity matching [36, 57], and depth cue [8, 9, 10]. Due to the limited discrimination of hand-crafted descriptors, these methods cannot well extract the high-level object semantics, leading to unsatisfactory performance in complex scenes. In recent years, learning based methods have achieved more competitive performance by generating more robust and representative co-saliency features, including un-/semi- supervised learning methods [22, 23, 31, 41, 53, 59], fully supervised methods based on Convolutional Neural Networks (CNNs) [26, 30, 38, 43, 46, 54], and Graph Convolutional Networks (GCNs) [27, 28, 55]. However, based on our observations, there are still three aspects of critical issues that need to be further considered and addressed in terms of modeling and exploiting inter-image relationships for co-saliency detection:

*(1) Insufficient group-wise relationship modeling.* Previous studies adopted the recurrent learning [30] and concatenation-convolution [46] strategies to model the inter-image relationships. However, the learned group representations vary with different order of the input group images, leading to unstable training and vulnerable inference. Thus, we propose a group-attentional semantic aggregation module to generate order-insensitive group semantics. Besides, in order to tackle the spatial variation of co-saliency patterns, we further consider long-range semantic dependencies across group images through a global attention mechanism when exploring the inter-image correspondences.

*(2) Competition between intra-image saliency and inter-image correspondence.* Despite the great efforts to boost the learning of group-wise representations, little attention is paid to the effective and reasonable group-individual combination. In existing works [30, 43], the learned group semantics were directly duplicated and concatenated with individual features. In fact, this operation implies that different individuals receive identical group semantics, which may propagate redundant and distracting information from the interactions among other images. To solve this issue, we design a gated group distribution module to learn dynamic combination weights between individual features and group-wise representations, thereby allowing different individuals to adaptively select useful group-wise information that is conducive to their own co-saliency prediction.

*(3) Weakened group consistency during feature decoding.* In the feature decoding of the CoSOD task, existing up-sampling or deconvolution based methods [54, 55] ignore the maintenance of inter-image consistency, which may lead to the inconsistency of co-salient objects among different images and introduce additional artifacts. Motivated by this problem, we develop a group consistency preserving decoder that further maintains inter-image relationships during the feature decoding process, predicting more consistent full-resolution co-saliency maps.

In this paper, we present an end-to-end collaborative aggregation-and-distribution network (CoADNet) for the CoSOD task, and the main contributions can be summarized as follows.

(1) The proposed CoADNet provides some insights and improvements in terms of modeling and exploiting inter-image relationships in the CoSOD workflow, and produces more accurate and consistent co-saliency results on four prevailing co-saliency benchmark datasets.

(2) We design an online intra-saliency guidance (OIaSG) module for supplying saliency prior knowledge, which is jointly optimized to generate trainable saliency guidance information. In this way, the network dynamically learns how to combine the saliency cues with deep individual features, which has better flexibility and expansibility in providing reliable saliency priors.

(3) We propose a two-stage aggregate-and-distribute architecture to learn group-wise correspondences and co-saliency features. In the first stage, a group-attentional semantic aggregation (GASA) module is proposed to model inter-image relationships with long-range semantic dependencies. In the second stage, we propose a gated group distribution (GGD) module to distribute the learned group semantics to different individuals in a dynamic and unique way.

(4) A group consistency preserving decoder (GCPD) is designed to replace conventional up-sampling or deconvolution driven feature decoding structures, which exploits more sufficient inter-image constraints to generate full-resolution co-saliency maps while maintaining group-wise consistency.

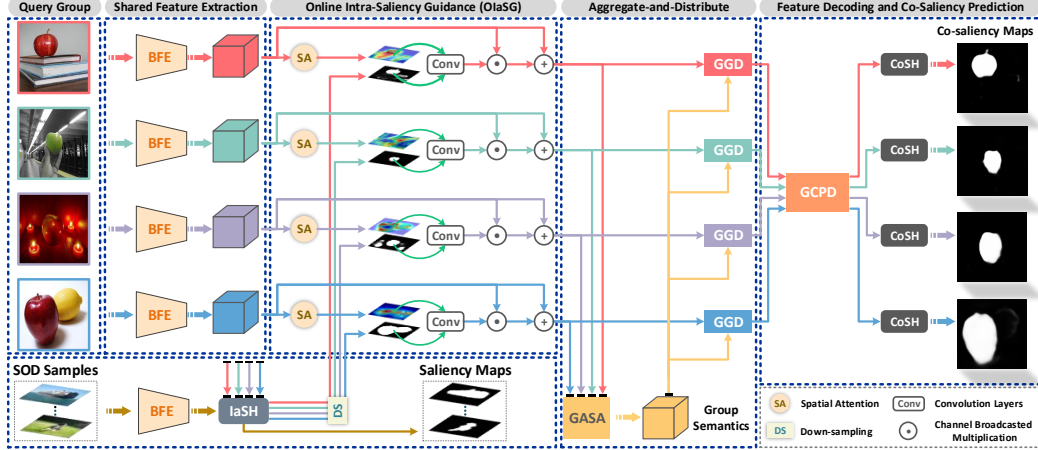

Figure 1: The flowchart of the proposed CoADNet. Given a query image group, we first obtain the deep features with a shared backbone feature extractor (BFE) and integrate learnable saliency priors with an OIaSG structure. The generated intra-saliency features are aggregated into group semantics through the group-attentional semantic aggregation (GASA) module, then they are further adaptively distributed to different individuals via the gated group distribution (GGD) module to learn co-saliency features. In the end, a group consistency-preserving decoder (GCPD) followed by a co-saliency head (CoSH) is used to predict consistent and full-resolution co-saliency maps.

## 2 Proposed Method

### 2.1 Overview

Given an input query group containing $N$ relevant images $\mathcal{I} = \{I^{(n)}\}_{n=1}^N$, the goal of CoSOD is to distinguish salient and repetitive objects from non-salient background as well as the salient but not commonly-shared objects, predicting the corresponding co-saliency maps. The pipeline of the proposed CoADNet is illustrated in Fig. 1. We start by extracting deep features through a shared backbone for the group-wise inputs. An online intra-saliency guidance (OIaSG) module is designed to mine intra-saliency cues that are further fused with individual image features. Then, a group-attentional semantic aggregation (GASA) module and a gated group distribution (GGD) module are integrated in a two-stage aggregate-and-distribute architecture to aggregate group semantic representations and adaptively distributes them to different individuals for co-saliency feature learning. Finally, low-resolution co-saliency features are fed into a group consistency preserving decoder (GCPD) followed by a co-saliency head to consistently highlight co-salient objects and produce full-resolution co-saliency maps.

### 2.2 Online Intra-Saliency Guidance

Conceptually, the CoSOD task can be decomposed into the two key components, *i.e.*, *saliency* and *repetitiveness*. The former is the cornerstone that focuses on visually-attractive regions of interest, and the latter constrains the objects repeatedly appearing in the group. However, the challenges of this task are that 1) the salient objects within an individual image may not occur in all the other group images, and 2) the repetitive patterns are not necessarily visually attractive, making it difficult to learn a unified representation to combine these two factors. Thus, we adopt a joint learning framework to provide trainable saliency priors as guidance information to suppress background redundancy.

Specifically, the group inputs are fed into the backbone network in a weight-sharing manner and embedded into a set of backbone features $F^{(n)} \in \mathbb{R}^{C \times H \times W}$. As shown in Fig. 1, we employ an intra-saliency head (IaSH) to infer online saliency maps, max-pooled as $E^{(n)} \in \mathbb{R}^{1 \times H \times W}$. Inspired by spatial attention mechanisms [48], we fuse online saliency priors with spatial feature attentions:

$$\widetilde{F}^{(n)} = \sigma(f^{3\times3}([F^{(n)}_{cap}; F^{(n)}_{cmp}])), \tag{1}$$

$$U^{(n)} = F^{(n)} + F^{(n)} \odot \sigma(f^{3\times3}[\widetilde{F}^{(n)}; E^{(n)}])), \tag{2}$$

where $F^{(n)}_{cap}$ and $F^{(n)}_{cmp}$ are derived by channel-wise average-pooling and max-pooling of $F^{(n)}$, respectively, square brackets represent channel concatenation, $f^{3\times3}$ denotes convolutions with

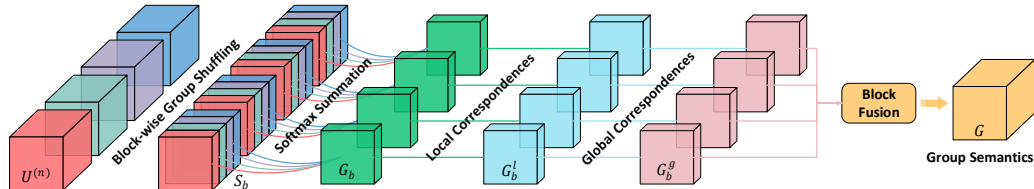

Figure 2: Illustration of the GASA module.

$3 \times 3$ filter size squeezing the inputs into single-channel maps, $\sigma$ represents the sigmoid activation, and $\odot$ means element-wise multiplication broadcasted along feature planes. In this way, we obtain a set of intra-saliency features (IaSFs) $\{U^{(n)}\}_{n=1}^{N}$ with suppressed background redundancy.

Benefiting from the joint optimization framework of intra-saliency branch, the co-saliency branch receives more reliable and flexible guidance. In the training phase, in addition to the group inputs loaded from a CoSOD dataset, we simultaneously feed $K$ auxiliary samples loaded from a SOD dataset into the shared feature backbone with the IaSH, generating single-image saliency maps $\mathcal{A} = \{A^{(k)}\}_{k=1}^{K}$. The saliency and co-saliency prediction is jointly optimized as a multi-task learning framework with better flexibility and expansibility in terms of providing reliable saliency priors .

## 2.3 Group-Attentional Semantic Aggregation

To efficiently capture discriminative and robust group-wise relationships, we investigate three key criteria: 1) *Insensitivity to input order* means that the learned group representations should be insensitive to the input order of group images; 2) *Robustness to spatial variation* considers the fact that co-salient objects may be located at different positions across images; 3) *Computational efficiency* takes the computation burden into account especially when processing large query groups or high-dimensional features. Accordingly, we propose a computation-efficient and order-insensitive group-attentional semantic aggregation (GASA) module which builds local and global associations of co-salient objects in group-wise semantic context. The detailed GASA module is shown in Fig. 2.

Directly concatenating the IaSFs to learn group relationships inevitably leads to high computational complexity and order-sensitivity. This motivates us to design a feature rearrangement strategy by adapting channel shuffling [56] to block-wise group shuffling that re-arranges the feature channels at the block level. Specifically, we first split each IaSF $U^{(n)}$ into $B$ feature blocks $\{U_b^{(n)}\}_{b=1}^{B}$ and sequentially concatenate the IaSFs into $U = [\{\{U_b^{(n)}\}_{b=1}^{B}\}_{n=1}^{N}]$. Through block-wise group shuffling, the concatenated group feature $U$ is transformed into $\widetilde{U} = [\{\{U_b^{(n)}\}_{n=1}^{N}\}_{b=1}^{B}]$, in which $S_b = [\{U_b^{(n)}\}_{n=1}^{N}]$ collects the $b^{th}$ feature blocks coming from all the $N$ IaSFs. To achieve permutation invariance, we first apply channel-wise softmax to the whole $S_b$, and then make a summation of all its $N$ features blocks. Repeating this operation on each $S_b$, we can obtain the corresponding block-wise group feature $G_b \in \mathbb{R}^{D \times H \times W}$ that uniformly encodes inter-image semantic information.

In order to capture richer semantics via jointly attending to information from different representation subspaces, each block-wise group feature $G_b$ is processed independently and then fused to formulate group semantic representations. Since $G_b$ only integrates inter-image features of the same positions, we further aggregate inter-image relationships among different spatial locations. Existing group aggregation methods only model local correspondences and cannot model long-range dependencies of scattered co-salient objects well, thus we encode the local and global associations in a group-wise attention architecture. We first exploit atrous convolutions with different dilation rates [4] to integrate multi-receptive-field features and capture local-context information. Specifically, the feature maps from different atrous convolutional layers are firstly concatenated and fed into a $1 \times 1$ convolutional layer for cross-channel communication. This procedure can be formulated as:

$$G_b^l = f^{1 \times 1}([\{f_k^{3 \times 3}(G_b)\}_{k=1,3,5,7}]), \tag{3}$$

where $f_k^{3 \times 3}$ denotes $3 \times 3$ convolution with dilation rate $k$ to generate a squeezed $D/4$ dimensional feature map, and $f^{1 \times 1}$ is a $1 \times 1$ convolutional layer for keeping input channel dimensions. In fact, this procedure builds inter-image associations of the same local regions.

Since co-salient objects may appear at any spatial locations across different query images, inspired by the self-attention mechanism [42], we model long-range semantic dependencies in an attention

based manner. For an aggregated feature block $G_b^l$, we generate three new features viewed as *query*, *key*, and *value* maps through parallel convolutional layers, which can be defined as:

$$G_b^{q/k/v} = \mathcal{R}(f^{q/k/v}(G_b^l)), \tag{4}$$

where $f^q$, $f^k$, and $f^v$ are three independent convolutional layers, $\mathcal{R}$ defines a reshaping operator that flattens the spatial dimension of input 3D tensors, leading to $G_b^{q/k/v} \in \mathbb{R}^{D \times (H \cdot W)}$. Then, we can construct the corresponding global attentional feature $G_b^g \in \mathbb{R}^{D \times H \times W}$ as:

$$G_b^g = \mathcal{R}^{-1}(G_b^v * CSM((T_r(G_b^q) * G_b^k)/\sqrt{D})) + G_b^l, \tag{5}$$

where $*$ is matrix multiplication, $T_r$ is matrix transposition, $CSM(\cdot)$ denotes column-wise softmax, and $\mathcal{R}^{-1}$ is the inverse operation of $\mathcal{R}$. Note that each block-wise group feature $G_b$ is transformed to the global attentional feature $G_b^g$ independently without weight-sharing. After that, we apply a $1 \times 1$ convolutional layer to $[\{G_b^g\}_{b=1}^B]$ for block fusion, obtaining the group semantics $G \in \mathbb{R}^{C \times H \times W}$.

## 2.4 Gated Group Distribution

In previous studies, the learned group-wise semantics are directly replicated and then concatenated with the intra-image features, which implies that group-wise information are equally exploited by different query images. In fact, the group-wise semantics encode the relationships of all images, which may include some distracting information redundancy for co-saliency prediction of different images. Hence, we propose a gated group distribution (GGD) module to adaptively distribute the most useful group-wise information to each individual. To achieve this, we construct a group importance estimator that learns dynamic weights to combine group semantics with different IaSFs through a gating mechanism. Specifically, we first concatenate $U^{(n)}$ and $G$, and then apply a $1 \times 1$ convolutional layer for channel reduction, producing $U_g^{(n)} \in \mathbb{R}^{C \times H \times W}$

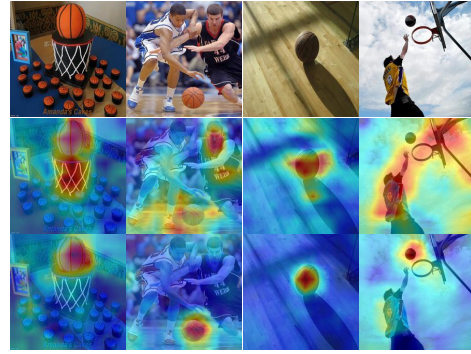

Figure 3: Visualizations of the GGD module.

as the input to the estimator. Then, we can obtain a probability map $P \in \mathbb{R}^{C \times H \times W}$ as follows:

$$P = \sigma(f^p(SE(U_g^{(n)}))), \tag{6}$$

where $SE$ is a squeeze-and-excitation block [24] implementing channel attention, and $f^p$ is a bottleneck convolutional layer. Intuitively, we consider $P$ as a probabilistic measurement that determines the linear combination weights between group semantics and intra-saliency features. Thus, the co-saliency features $X^{(n)}$ can be derived by the gated operation:

$$X^{(n)} = P \otimes G + (1 - P) \otimes U^{(n)}, \tag{7}$$

where $\otimes$ denotes Hadamard product. Note that the GGD module is shared for all the IaSF inputs. Some examples are visualized in Fig. 3. We can see that the IaSFs (the second row) contain much redundant information, while the GGD module generates co-saliency features (the third row) that better highlight the co-salient areas (*i.e.*, basketball).

## 2.5 Group Consistency Preserving Decoder

The hierarchical feature abstraction framework usually produces low-resolution deep features, which should be up-scaled to generate full-resolution prediction results. However, common up-sampling- or deconvolution-based feature decoders are not suitable for the CoSOD task because they ignore the valuable inter-image constraints and may weaken the group consistency during the prediction process. To tackle this issue, we propose a group consistency preserving decoder (GCPD) to consistently predict the full-resolution co-saliency maps.

The GCPD is composed of three cascaded feature decoding (FD) units, through each of which the feature resolution is doubled while the feature channels are halved. The details of the FD unit are illustrated in Fig. 4. In each unit, the input co-saliency features $X^{(n)} \in \mathbb{R}^{C \times H \times W}$ are transformed to $\widehat{X}^{(n)} \in \mathbb{R}^{C_d \times 2H \times 2W}$ via a $1 \times 1$ convolution and a $2\times$ deconvolution, where $C_d = C/2$. After that, we apply global average pooling (GAP) to $\widehat{X}^{(n)}$ and obtain $N$ vectorized representations $\widehat{x}^{(n)} \in \mathbb{R}^{C_d}$, which are further arranged into the rows of matrix $Y \in \mathbb{R}^{N \times C_d}$.

Next, we apply the column-wise softmax and the row-wise summation to $Y$, producing a compact group-wise feature vector $y \in \mathbb{R}^{C_d}$. Note that this aggregation strategy is also order-insensitive. Thus, the output higher-resolution feature maps can be deduced by:

$$X_{up2}^{(n)} = \widehat{X}^{(n)} \odot MLP([\widehat{x}^{(n)}; y]), \quad (8)$$

where $MLP$ is a shared multi-layer perceptron, which maps the concatenation of $\widehat{x}^{(n)}$ and $y$ into $C_d$ dimension. By stacking three cascaded FD units, we can obtain the $N$ decoded features $Z^{(n)} \in \mathbb{R}^{C/8 \times 8H \times 8W}$ with the finest spatial

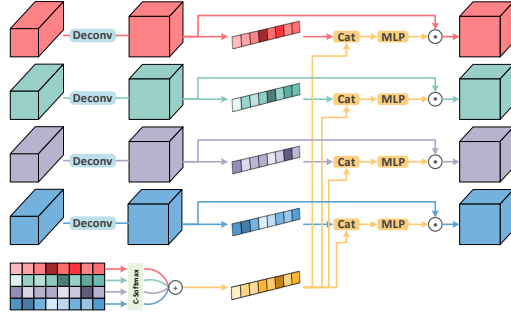

Figure 4: Illustration of each FD unit in GCPD.

resolution, which are further fed into a shared co-saliency head (CoSH) to generate full-resolution co-saliency maps $M^{(n)}$. Here, the CoSH includes a $1 \times 1$ convolutional layer equipped with the sigmoid activation.

## 2.6 Supervisions

We jointly optimize the co-saliency and single image saliency predictions in a multi-task learning framework. Given $N$ co-saliency maps and ground-truth masks (*i.e.*, $\{M^{(n)}\}_{n=1}^N$ and $\{T_c^{(n)}\}_{n=1}^N$), and $K$ auxiliary saliency predictions and ground-truth masks (*i.e.*, $\{A^{(k)}\}_{k=1}^K$ and $\{T_s^{(k)}\}_{k=1}^K$), we build a joint objective function $\mathcal{L}$ of the overall CoSOD pipeline via two binary cross-entropy losses:

$$\mathcal{L} = \alpha \cdot \mathcal{L}_c + \beta \cdot \mathcal{L}_s, \quad (9)$$

where $\mathcal{L}_c = -(\sum_{n=1}^N (T_c^{(n)} \cdot log(M^{(n)}) + (1 - T_c^{(n)})] \cdot log(1 - M^{(n)})))/N$ is the co-saliency loss, and $\mathcal{L}_s = -(\sum_{k=1}^K (T_s^{(k)} \cdot log(A^{(k)}) + (1 - T_s^{(k)}) \cdot log(1 - A^{(k)})))/K$ is the auxiliary saliency loss.

## 3 Experiments

### 3.1 Benchmark Datasets and Evaluation Metrics

In experiments, we conduct extensive evaluations on four popular datasets, including CoSOD3k [15], Cosal2015 [53], MSRC [47], and iCoseg [2]. *The detailed introduction can be found in the Supplemental Materials.* For quantitative evaluation, we use the Precision-Recall (P-R) curve, F-measure [1], MAE score [45], and S-measure [13]. The P-R curve visually shows the variation tendency between different precision and recall scores in a curve fashion. The closer the curve is to (1,1), the better the algorithm performance. F-measure is a comprehensive measurement of precision and recall scores, with a larger value indicating a better performance. MAE score calculates the difference between the continuous saliency map and ground truth, and a smaller value correspond to better performance. S-measure score evaluates the structural similarity between the saliency map and ground truth, with a larger value representing a better performance.

### 3.2 Implementation Details

Following the common development protocols as adopted in [46, 43, 55], we sequentially divide each input group into a series of sub-groups consisting of 5 images in a non-overlapping manner. For the last sub-group with images less than 5, we supplement by randomly selecting samples from the whole query group. In each training iteration, 24 sub-groups from the co-saliency dataset COCO-SEG [43] and 64 samples from the saliency dataset DUTS [44] are simultaneously fed into the network for jointly optimizing the objective function in Eq. 9, where $\alpha = 0.7$ and $\beta = 0.3$, by the Adam [29] algorithm with a weight decay of $5e^{-4}$. In our experiments, we evaluate three commonly used backbone networks, including VGG16 [39], ResNet-50 [21], and Dilated ResNet-50 [51]. To enlarge the spatial resolution of the final convolutional feature maps, minimal modifications are made to the original backbone networks to achieve an overall down-sampling ratio of 8. For VGG16 [39], we remove the first and the last max-pooling layer. For ResNet50 [21], we remove the max-pooling layer and change the stride of the first convolutional layer from (2,2) to (1,1). Considering the different embedding dimension of backbone feature maps, in block-wise group-shuffling, the factor $B$ is set to 8 for ResNet-50 [21] and Dilated ResNet-50 [51], and decreased to 4 for VGG16 [39]. For

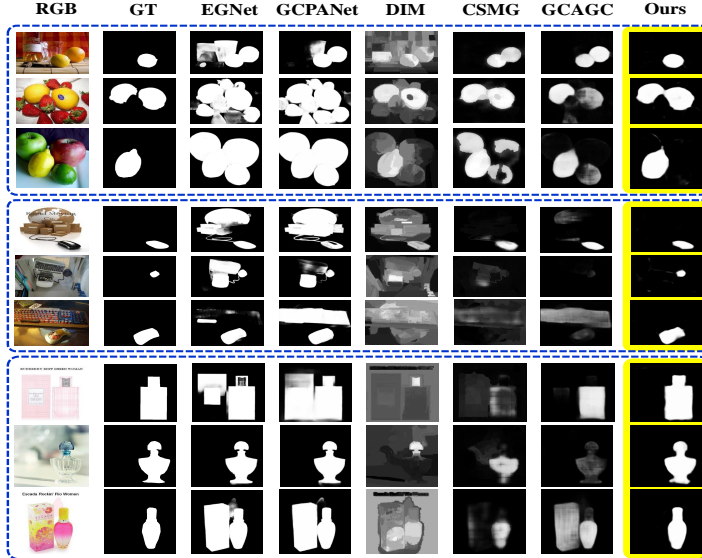

Figure 5: Visual examples of co-saliency maps generated by different methods. Our method with the Dilated ResNet-50 backbone is marked with the yellow shading.

convenience, all input images are uniformly resized to $224 \times 224$, and restored to the original size for testing. The proposed framework is implemented in MindSpore and accelerated by $4$ Tesla P100 GPUs [3]. In practice, we set the initial learning rate to $1e^{-4}$ that is halved every $5,000$ iterations, and the whole training process converges until $50,000$ iterations. The average inference time for a single image is $0.07$ second.

## 3.3 Comparisons with State-of-the-arts

To verify the effectiveness of the proposed method, we compare it with ten state-of-the-art methods on four datasets, including three latest deep learning-based SOD methods for single image (*i.e.*, GCPANet [5], CPD [49], and EGNet [58]), and seven learning based CoSOD methods (*i.e.*, UMLF [20], GoNet [23], DIM [25], RCGS [43], CODW [53], CSMG [54], and GCAGC [55]). All the results are provided by the authors directly or generated by the source codes under the default parameter settings in the corresponding models. *For the limited space, more details including visual examples, P-R curves, and ablation studies can be found in the Supplemental Materials.*

In Fig. 5, we provide some challenging examples in three different image groups (*i.e.*, lemon, mouse, and cosmetic bottle). Compared with other methods, our method achieves more competitive visual performance, mainly in the following aspects: *(a) More accurate co-salient object localization.* Our method can detect the co-salient object more completely and accurately. For example, in the second group, only our method can accurately predict the co-salient object in each image, even if the co-salient object is small or shares the similar color appearance with backgrounds. Moreover, our method shows superiority in terms of the internal consistency and structural integrity. *(b) Stronger background suppression.* Our method can address the complex background interference, such as the non-salient background regions or salient but non-repetitive regions. For example, in the first image group, the orange in the first image is salient in the individual image, but not appears repeatedly. For this case, only our method can effectively highlight the co-salient object and suppress the interference. *(c) More effective inter-image relationship modeling ability.* Our method has more accurate and robust correspondence modeling capability. For example, in the third image group, only the cosmetic bottle is salient and repetitive object. However, it is difficult to accurately capture such high-level semantic correspondence information through simple appearance pattern learning. Thus, except for the proposed method, all compared methods fail to correctly detect the co-salient object.

For quantitative evaluations, we report three popular metrics of F-measure, MAE score, and S-measure in Table 1. Note that no matter which backbone network is applied, our framework achieves the best performance in all measures across all datasets. Taking the most powerful variant CoADNet-DR as

Table 1: Quantitative comparisons over four benchmark datasets. The bold numbers indicate the best performance on the corresponding dataset. '-V', '-R' and '-DR' mean the VGG16 [39], ResNet-50 [21], and Dilated ResNet-50 [51] backbones, respectively.

| | Cosal2015 Dataset | | | CoSOD3k Dataset | | | MSRC Dataset | | | iCoseg Dataset | | |
|---|---|---|---|---|---|---|---|---|---|---|---|---|
| | $F_\beta \uparrow$ | MAE↓ | $S_m \uparrow$ | $F_\beta \uparrow$ | MAE↓ | $S_m \uparrow$ | $F_\beta \uparrow$ | MAE↓ | $S_m \uparrow$ | $F_\beta \uparrow$ | MAE↓ | $S_m \uparrow$ |
| CPD [49] | 0.8228 | 0.0976 | 0.8168 | 0.7661 | 0.1068 | 0.7788 | 0.8250 | 0.1714 | 0.7184 | 0.8768 | 0.0579 | 0.8565 |
| EGNet [58] | 0.8281 | 0.0987 | 0.8206 | 0.7692 | 0.1061 | 0.7844 | 0.8101 | 0.1848 | 0.7056 | 0.8880 | 0.0601 | 0.8694 |
| GCPANet [5] | 0.8557 | 0.0813 | 0.8504 | 0.7808 | 0.1035 | 0.7954 | 0.8133 | 0.1487 | 0.7575 | 0.8924 | 0.0468 | 0.8811 |
| UMLF [20] | 0.7298 | 0.2691 | 0.6649 | 0.6895 | 0.2774 | 0.6414 | 0.8605 | 0.1815 | 0.8007 | 0.7623 | 0.2389 | 0.6828 |
| CODW [53] | 0.7252 | 0.2741 | 0.6501 | – | – | – | 0.8020 | 0.2645 | 0.7152 | 0.8271 | 0.1782 | 0.7510 |
| DIM [25] | 0.6363 | 0.3126 | 0.5943 | 0.5603 | 0.3267 | 0.5615 | 0.7419 | 0.3101 | 0.6579 | 0.8273 | 0.1739 | 0.7594 |
| GoNet [23] | 0.7818 | 0.1593 | 0.7543 | – | – | – | 0.8598 | 0.1779 | 0.7981 | 0.8653 | 0.1182 | 0.8221 |
| CSMG [54] | 0.8340 | 0.1309 | 0.7757 | 0.7641 | 0.1478 | 0.7272 | 0.8609 | 0.1892 | 0.7257 | 0.8660 | 0.1050 | 0.8122 |
| RCGS [43] | 0.8245 | 0.1004 | 0.7958 | – | – | – | 0.7692 | 0.2134 | 0.6717 | 0.8005 | 0.0976 | 0.7860 |
| GCAGC [55] | 0.8666 | 0.0791 | 0.8433 | 0.8066 | 0.0916 | 0.7983 | 0.7903 | 0.2072 | 0.6768 | 0.8823 | 0.0773 | 0.8606 |
| CoADNet-V | 0.8748 | 0.0644 | 0.8612 | 0.8249 | 0.0696 | 0.8368 | 0.8597 | 0.1139 | 0.8082 | 0.8940 | 0.0416 | 0.8839 |
| CoADNet-R | 0.8771 | 0.0609 | 0.8672 | 0.8204 | **0.0643** | 0.8402 | **0.8710** | **0.1094** | **0.8269** | 0.8997 | 0.0411 | 0.8863 |
| CoADNet-DR | **0.8874** | **0.0599** | **0.8705** | **0.8308** | 0.0652 | **0.8416** | 0.8618 | 0.1323 | 0.8103 | **0.9225** | 0.0438 | **0.8942** |

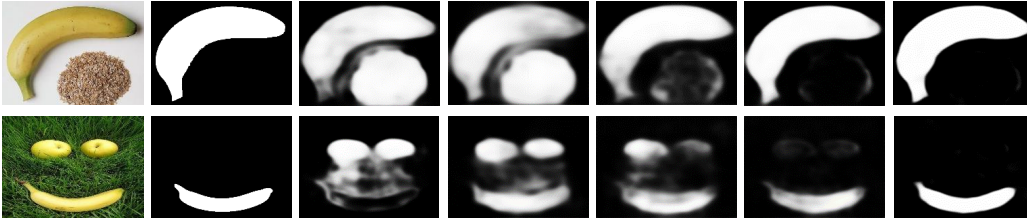

Figure 6: Visualization of different ablative results. From left to right: Input image group, Ground truth, Co-saliency maps produced by the Baseline, Baseline+OIaSG, Baseline+OIaSG+GASA, Baseline+OIaSG+GASA+GGD, and the full CoADNet.

Table 2: Quantitative evaluation of ablation studies on the Cosal2015 and CoSOD3k datasets.

| Modules | | | | | Cosal2015 Dataset | | | CoSOD3k Dataset | | |
|---|---|---|---|---|---|---|---|---|---|---|
| Baseline | OIaSG | GASA | GGD | GCPD | $F_\beta \uparrow$ | MAE↓ | $S_m \uparrow$ | $F_\beta \uparrow$ | MAE↓ | $S_m \uparrow$ |
| ✓ | | | | | 0.7402 | 0.1406 | 0.7459 | 0.7099 | 0.1170 | 0.7320 |
| ✓ | ✓ | | | | 0.8023 | 0.1161 | 0.7967 | 0.7489 | 0.1138 | 0.7721 |
| ✓ | ✓ | ✓ | | | 0.8465 | 0.0946 | 0.8209 | 0.8008 | 0.0915 | 0.8089 |
| ✓ | ✓ | ✓ | ✓ | | 0.8682 | 0.0712 | 0.8534 | 0.8211 | 0.0815 | 0.8223 |
| ✓ | ✓ | ✓ | ✓ | ✓ | **0.8874** | **0.0599** | **0.8705** | **0.8308** | **0.0652** | **0.8416** |

an example, compared with the **second best competitor** on the Cosal2015 dataset, the percentage gain reaches 2.4% for F-measure, 24.3% for MAE score, and 2.4% for S-measure, respectively. On the CoSOD3k dataset, the **minimum percentage gain** against the comparison methods reaches 3.0% for F-measure, 28.8% for MAE score, and 5.4% for S-measure, respectively. Interestingly, we found that the stronger backbone did not achieve the best performance on the smaller MSRC dataset, while it still shows more competitive performance on the other three larger datasets. It is worth mentioning that in our comparison experiments the second best method GCAGC [55] is established upon the VGG16 [39] backbone network, containing 119 MB parameters totally. The proposed CoADNet-V shares a very close number of parameters (121 MB), which is adequate for proving the superiority of our solution under similar model capacity.

## 3.4 Ablation Studies

To verify our contributions, we design different variants of our CoADNet with the Dilated ResNet-50 backbone by removing or replacing the four key modules (*i.e.*, OIaSG, GASA, GGD, and GCPD). To construct our baseline model, we simplified the CoADNet as follows: 1) removing the OIaSG module, and pretraining the feature backbone on the same SOD dataset DUTS [44]; 2) replacing the GASA module with standard $3 \times 3$ convolutions; 3) replacing the GGD module with direct concatenation followed by a $1 \times 1$ convolution; 4) replacing GCPD with three cascaded deconvoluton layers. The

Table 3: Detection performance of our CoADNet-V using CoSOD3k as the training set.

| | Cosal2015 Dataset | | | MSRC Dataset | | | iCoseg Dataset | | |
|---|---|---|---|---|---|---|---|---|---|
| | $F_\beta \uparrow$ | MAE $\downarrow$ | $S_m \uparrow$ | $F_\beta \uparrow$ | MAE $\downarrow$ | $S_m \uparrow$ | $F_\beta \uparrow$ | MAE $\downarrow$ | $S_m \uparrow$ |
| CoADNet-V | 0.8592 | 0.0818 | 0.8454 | 0.8347 | 0.1558 | 0.7670 | 0.8784 | 0.0725 | 0.8569 |

baseline model is carefully designed to share a very similar parameter number with the full model. For clarity, the architecture of the baseline model is further illustrated in the *Supplemental Materials*. We gradually add the four key modules to the baseline, and provide the visualization and quantitative results in Fig. 6 and Table 2.

In Fig. 6, it is observed that the baseline model can roughly locate the salient object, but fails to suppress the non-common salient object and background (*e.g.*, the cereal and apple). By introducing the the OIaSG structure that provides saliency prior information, the salient object (*e.g.*, the banana in the second image) is highlighted and the background regions (*e.g.*, the grass) are effectively suppressed, thereby promoting the percentage gain of F-measure reaches $8.4\%$ on Cosal2015 and $5.5\%$ on CoSOD3k. Then, introducing the GASA module that learns more discriminative group semantic representations further suppresses the non-common salient objects (*e.g.*, the cereal and apple), and boosts the performance with large margins. For example, on the CoSOD3k benchmark, the F-measure, MAE, and S-measure gained a relative improvement of $6.9\%$, $19.6\%$, and $4.8\%$, respectively. Subsequently, the GGD module contributes to $2.6\%$ and $4.0\%$ gains in terms of F-measure and S-measure on Cosal2015 dataset by dynamically distributing the group semantics to different individuals. Finally, with our GCPD, the co-salient object is highlighted more consistently and completely, which further boosts the whole framework to state-of-the-art on all datasets.

In addition to the component analysis, we also evaluate the detection performance of the CoADNet-V model trained on the CoSOD3k [15] dataset. Previous state-of-the-art deep co-saliency detectors employ the large-scale COCO-SEG [43] dataset that sufficiently contains more than $200,000$ image samples for model training. Despite the abundant annotated data, COCO-SEG [43] is modified from the original semantic segmentation benchmark MS COCO [37] following simple pre-defined selection strategies. Consequently, a considerable proportion of the co-occurring target objects within query groups do not strictly meet the requirement of being *salient*, which inevitably propagates ambiguous supervision information to the trained co-saliency models. The recently released CoSOD3k [15] dataset is particularly tailored for the co-salient object detection task. Although it is a much smaller dataset containing 3316 images, it better suits the definition of *co-saliency*. Hence, we provide the detection results of our CoADNet-V variant in Table 3, which also indicate competitive performance. In future researches, we will continue to explore the possibility of training the model with the more appropriate CoSOD3k [15] dataset.

## 4   Conclusion

We proposed an end-to-end CoSOD network by investigating how to model and utilize the inter-image correspondences. We first decoupled the single-image SOD from the CoSOD task and proposed an OIaSG module to provide learnable saliency prior guidance. Then, the GASA and GGD modules are integrated into a two-stage aggregate-and-distribute structure for effective extraction and adaptive distribution of group semantics. Finally, we designed a GCPD structure to strengthen inter-image constraints and predict full-resolution co-saliency maps. Experimental results and ablative studies demonstrated the superiority of the proposed CoADNet and the effectiveness of each component.

## Acknowledgements

This work was completed by Qijian Zhang as an intern at the Institute of Information Science, Beijing Jiaotong University. This work was supported in part by the Beijing Nova Program under Grant Z201100006820016, in part by the National Key Research and Development of China under Grant 2018AAA0102100, in part by the National Natural Science Foundation of China under Grant 62002014, Grant 61532005, Grant U1936212, Grant 61972188, in part by the Hong Kong RGC under Grant 9048123 (CityU 21211518), Grant 9042820 (CityU 11219019), in part by the Fundamental Research Funds for the Central Universities under Grant 2019RC039, in part by Hong Kong Scholars Program, in part by Elite Scientist Sponsorship Program by the Beijing Association for Science and Technology, in part by CAAI-Huawei MindSpore Open Fund, and in part by China Postdoctoral Science Foundation under Grant 2020T130050, Grant 2019M660438.

## Broader Impact

This work aims at general theoretical issues for the co-salient object detection problem and does not present any foreseeable societal consequence.

## Footnotes

†Corresponding author: Runmin Cong (rmcong@bjtu.edu.cn)

[3]We also provide a Pytorch implementation version

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
