[Supplementary Material]

# CoADNet: Collaborative Aggregation-and-Distribution Networks for Co-Salient Object Detection (*Supplementary Material*)

## 1 Details of Benchmark Datasets

We conduct extensive experiments on four prevailing CoSOD benchmark datasets, including MSRC [4], iCoseg [1], Cosal2015 [6], and CoSOD3k [2]. MSRC [4] is the first available dataset for co-saliency detection, which includes 7 groups of 210 images after removing the non-salient *grass* group, and each group contains 30 images. iCoseg [1] is originally designed for image co-segmentation but is also widely used for co-saliency detection, which consists of 643 images distributed in 38 groups. The group size varies from 4 to 42. In this dataset, the co-salient objects within one group usually share similar background context and object appearances with variation of sizes and poses, and thus is viewed as a relatively simple dataset. Cosal2015 [6] is the first large-scale dataset for co-saliency detection and contains 2,015 images from 50 groups, and each group including 26~52 images. This dataset is much more difficult, and is characterized by complex contextual information, occusion, large appearance variation, and background clutters. CoSOD3k [2] is the latest and largest dataset carefully tailored for the task of co-salient object detection, which consists of 3,316 images divided in 160 groups. Compared with the other three CoSOD benchmark datasets, CoSOD3k makes a significant leap in terms of sufficient group semantic categories, diverse object sizes and appearances. Hence, it is considered as the most challenging CoSOD benchmark dataset.

## 2 Experiments

In this section, we provide more visual examples (*i.e.*, the banana, popsicle, pepper, and bowl groups) in Fig. 1, the P-R curves of different competing methods on four datesets in Fig. 2, and the detailed flowchart of baseline model is shown in Fig. 3.

From Fig. 1, we can observe that the proposed CoADNet produces more visually competitive results with accurate localization, complete structure, and high consistency of co-salient objects. Moreover, our method can effectively suppress the cluttered background regions. As shown in Fig. 2, we can see that our method with the Dilated ResNet-50 [5] backbone (*i.e.*, the red solid line) achieves the highest precision on all datasets except the smallest MSRC dataset. Overall, both qualitative and quantitative results in the main body and supplementary material demonstrate the superiority and effectiveness of our method.

For clarity, in this *Supplementary Material*, we also intuitively provide the structure of the basline model in Fig. 3. Note that, for fair comparison, the backbone feature extractor in baseline model is pretrained on a single-image saliency detection dataset (*i.e.*, DUTS [3]) for shared feature learning.

Figure 1: Visual examples of co-saliency maps generated by different methods. Our method with the Dilated ResNet-50 backbone is marked with the yellow shading.

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

Figure 3: The overall framework of our baseline model constructed for ablative analysis. The input group of images pass through the backbone feature extractor pretrained (P-BFE) on a single-image saliency detection dataset for shared feature learning. The generated individual image features are sequentially concatenated and convolved into group semantic representations, which are further equally concatenated back with different individuals to obtain co-saliency features. During feature decoding, we apply three cascaded deconvolutional layers to produce full-resolution co-saliency features, which are fed into a co-saliency head (CoSH) to predict co-saliency maps.