[Reviews · NeurIPS 2020]

Review 1

Summary and Contributions: In this paper, a novel collaborative aggregation-and-distribution network is presented to detect the co-salient objects from an image group. The main contributions are three-fold, which attend to several major issues in the task of co-salient object detection. First, an OIaSG is constructed to combine saliency detection with co-saliency detection in a unified joint-optimization framework. Second, the paper proposed a two-step aggregation-and-distribution strategy to mine more discriminative group-wise information by GASA and adaptively capture co-saliency cues by GGD. Third, the authors strengthen the group image relationships in the process of feature up-sampling. The experiments show competitive performances both qualitatively and quantitatively. After reading rebuttal information. ========================== The authors clearly addressed my comments. I have also read other reviewers’ comments as well as the corresponding responses. The additional experiments and analyses seem to be convincing. I think the paper is interesting, which may be of interest for the researchers in the related areas.

Strengths: This paper investigated the critical problems of co-saliency object detection, including the input order issue, inter-image modeling issue, intra and inter information integration issue, and group consistency issue. The claims made by the authors are intuitive and turn to be effective. A novel GASA module was constructed for capturing robust group semantics, which remains unchanged facing varying input orders. The design of GASA also followed multi-head attention and included non-local aggregations to learn more powerful group-wise features. The GGD and GCPD further provided new ways of designing better deep architecture for the co-saliency detection task in terms of fully modeling inter-image interactions across all stages. The ideas of this paper are novel and insightful, the flowchart is reasonable and reproducible, and the experiments are sufficient and competitive.

Weaknesses: For the CoSOD task, some related images in an image group are used as the inputs of the model, which is similar to the video SOD task. Could you directly transform the CoADNet or some parts to the video SOD task? The authors claim that the jointly-optimized OIaSG module could provide more reliable saliency priors. Traditionally, a typical practice would be to pretrain the backbone network on SOD tasks to gain useful priors. However, it seems that the authors did not analyze why the proposed OIaSG could be a better choice. In other words, its effectiveness is not fully evaluated. Some of the experimental setups are not detailed enough. For example, the network takes five images as a sub-group, which means that it is often impossible to process all images from a certain category. In such context, how the images are sampled and exploited should be clearly explained. In the experiments, only the inferencing time is provided. How about the model size of the proposed method compared to the SOTA model, such as GCAGC? Will you release the source code in the future?

Correctness: yes

Clarity: yes

Relation to Prior Work: yes

Reproducibility: Yes

Additional Feedback:


Review 2

Summary and Contributions: This paper presents an end-to-end collaborative aggregation-and-distribution network (CoADNet) to capture both salient and repetitive visual patterns from multiple images. The proposed model contains an online intra-saliency guidance (OIaSG) module for supplying saliency prior knowledge, a two-stage aggregate-and-distribute architecture to learn group-wise correspondences and co-saliency features, and a group consistency preserving decoder (GCPD) to replace conventional up-sampling or deconvolution driven feature decoding structures. ---------after rebuttal------- 1) I have read the response and other reviewers' comments. While most minor concerns are addressed by authors, I still feel this paper is weak in high level insights and novelty. In my view, this work seems like an integrated version of most recent works, including group shuffling (simply applying channel shuffle [45] to multiple group of features) and group attention (multi-head attention). 2) Moreover, Table. 2 shows that with OIaSG, the model achieves significant performance boost (Line 285-288: 8.4% on Cosal2015 and 5.5% on CoSOD3k performance improvements according to F-measure). Yet, please also note that OIaSG requires additional labeled data for training, which are acquired from SOD dataset (the largest SOD dataset: DUTS). Therefore, it is not clear if the performance gains are from additional data and, if it is fair to other compared SOTAs.

Strengths: -The authors present comprehensive analysis about current issues of co-salient object detection. -The proposed method achieves state-of-the-art results on Cosal2015, CoSOD3k, MSRC and iCoseg datasets. -The paper is easy to read and follow.

Weaknesses: -The overall motivation of this work is not clearly stated after reading introduction. It is like an integrated model that exploits several incremental techniques to improve the performance. -The section of related work is missing. The authors should summarize the typical strategies about co-salient object detection, and some relevant modules in most recent works. -The experiments are not convincing. The presented results seem cannot support the claimed contributions, it's very hard to say what kind of feature is learned with the proposed method. The authors should provide some feature visualizations to verify the mentioned benefits. -There is a complete lack of discussing the impact of adding additional parameters and additional computational effort due to several modules appended on the baseline. The authors should provide this analysis for a fair comparison with other works. -In addition, this work is weakly relevant to the NeurIPS community.

Correctness: This paper is technically sound.

Clarity: This paper is generally well written.

Relation to Prior Work: Without related work. Not convincing.

Reproducibility: Yes

Additional Feedback: after reading the rebuttal and reviews from other reviewers. I still feel the overall design is too straight-forward and does not seem to be very interesting/exciting. Therefore, I can not vote for acceptance,


Review 3

Summary and Contributions: This paper proposes an end-to-end CoADNet framework for co-salient object detection (CoSOD), which produces state-of-the-art performance on four popular benchmark datasets. The key components of the proposed CoADNet include aggregating group-wise semantic representations and distributing the learned group semantics to individual image features. It uses the OIaSG, GASA, GCD, and GCPD modules to address the following issues: 1) how to bridge the relationship between saliency and co-saliency; 2) how to generate more discriminative group-wise features that are insensitive to the input order; 3) how to distribute group semantics to different individuals; 4) how to maintain inter-image constraints during feature decoding stage.

Strengths: +The authors discussed several important issues, which is not considered or well addressed in previous studies, in my opinion, is very inspiring for the CoSOD tasks. For example, an intuitive idea about group feature learning is that it must be order-invariant, and the GASA module elegantly solves it and also keeps efficient. +Another interesting idea is that the learned group representations should be adaptively distributed, instead of being directly duplicated for concatenation. +Previous CoSOD works do not consider whether it is necessary to consider co-saliency constraints during feature decoding, and this paper shows that it is also practical to further boost the performance by tailoring specific decoders for co-saliency cues mining.

Weaknesses: -The designed CoSOD framework only accepts fixed-number (i.e. 5 images) group-wise inputs at each time. How to handle the case during testing where the number of samples in a particular category is a non-integral multiple of 5? Is there any overlap between the data? -In the GASA module, there is a pre-defined parameter “b” which controls the number of partitioned feature blocks. However, the authors did not explain how to choose an appropriate value for “b” and to what extent will it influence the model. Besides, in the second paragraph of Sec 2.3, how to perform “block-wise group shuffling” is vague. More detailed descriptions are needed. -In the CoADNet framework, authors have repeatedly emphasized that the model is not sensitive to the input order, but has not specifically explained how it works. -The proposed CoADNet uses ResNet50 and Dilated ResNet50 as backbone networks. However, some of the competing methods only use less powerful backbone such as VGG-16. For fair comparison, the authors should also provide the experimental results with VGG-16 as backbone network.

Correctness: Most claims look correct to me.

Clarity: Yes, the paper is well written and easy to follow

Relation to Prior Work: The main difference between this work and the exisiting works is clearly discussed but some existing literatures on COSOD are mising.

Reproducibility: Yes

Additional Feedback: ---------------------------update after rebuttal---------------------------- In the authors’ feedback, the authors addressed my concerns on technical details and experimental comparisons. They implemented experiments to use VGG-16 as their backbone and the obtained performances are better than other state-of-the-art methods. Overall, the novelty, technicality, and performance of this submission are good to me. So I would like to keep my initial score and favor accepting it.


Review 4

Summary and Contributions: The authors proposed a network consisting of different variants to solve the task of Co-Salient object detection, which leads to improved performance. More specifically, the proposed pipeline integrates saliency priors, group-attentional semantic aggregation to achieve superior performance. Experiments on different datasets under various settings demonstrate the effectiveness of the proposed approach.

Strengths: 1. The combination of different components has shown to perform well to solve the task of co-saliency detection. 2. Joint optimization of intra-saliency guidance and the co-saliency query group helps to suppress the background information. 3. The paper is relatively well written and easy to follow. 4. A reasonable ablation study is provided to justify different variants used in the proposed method.

Weaknesses: 1. While the idea of integrating saliency priors, group-attentional semantic aggregation modules for Co-Saliency object detection is interesting, there is a complete lack of discussing the impact of adding additional parameters and computational effort due to the different variants used in the proposed complex pipeline. The authors should provide this analysis for a fair comparison with the closest baseline [44]. 2. I was wondering why the numbers reported in Table 1 for [44] are different from the results reported in [44]. Is it because [44] used VGG-16 as a backbone network while the authors used ResNet50 as the backbone. I am confused why the authors did not add a column to include the backbone network for each baseline method. For a fair comparison, I strongly suggest authors should experiment using the VGG-16 backbone to demonstrate superiority over the baselines. It is not clear if authors implemented baseline methods e.g. [44] using ResNet50 backbone. To show the superiority of the proposed method, I strongly believe that authors should report results using the VGG-16 backbone rather than implementing baselines using ResNet-50 backbone (if the authors used ResNet-50 to report baseline numbers). 3. More importantly, the idea of group semantics for co-saliency detection has been already explored in [32]. 4. Additionally, I was wondering why the authors did not report results using the AP metric. Is there any specific reason? 5. The idea of incorporating saliency priors to suppress the background information is nothing new in the context of saliency detection, semantic segmentation.

Correctness: The authors should clearly mention what backbone network each baseline method use for a fair comparison.

Clarity: Decent

Relation to Prior Work: Not strong enough.

Reproducibility: Yes

Additional Feedback: While the idea of combining different components is interesting in the context of co-saliency detection, it is hard to judge whether the proposed approach indeed demonstrates state-of-the-art results without the help of a stronger backbone network. The method outperforms the baselines probably due to the ResNet-50 backbone while other baselines methods used VGG based backbones. The reported numbers in experiments suggest noticeable improvements; however, there are several major issues mentioned in the weakness section that need to be explained before the final rating. After reading rebuttal information. ========================== While I am a bit more positive about the paper after reading the rebuttal and other reviews, my original concern regarding the technical novelty and contributions persist. The proposed model exploits several incremental techniques (group shuffling and group attention) to improve the overall performance as agreed on by R2. So, I am not convinced the paper has enough novelty to be presented as a new contribution to the NeurIPS community.

[Author Response · NeurIPS 2020]

Thanks for all reviewers' valuable comments. We will first answer the common questions then respond to each reviewer.

**[CQ1] Sub-group sampling (R1-Q3, R3-Q1):** Following the common testing protocol as adopted in Ref [44], we sequentially divide each input group into sub-groups consisting of 5 images in a non-overlapping manner. For the last sub-group with images less than 5, we supplement by randomly selecting samples from the whole given group.

**[CQ2] Model size & VGG16 Backbone (R1-Q4, R3-Q4):** The performance of our method with VGG16 backbone is shown in the table. 1) Our method can still achieve better performance than Ref [44]. 2) The model size of Ours-V is comparable to the method [44] (121 MB vs 119 MB). Since most CoSOD competitors did not release codes, here we only report the model size of [44] for comparison, which is provided directly by its authors. Our code will be released.

| | Cosal2015 | | | | CoSOD3k | | | | MSRC | | | | iCoseg | | | |
|---|---|---|---|---|---|---|---|---|---|---|---|---|---|---|---|---|
| | AP | $F_\beta$ | MAE | $S_m$ | AP | $F_\beta$ | MAE | $S_m$ | AP | $F_\beta$ | MAE | $S_m$ | AP | $F_\beta$ | MAE | $S_m$ |
| Ref [44] | 0.8846 | 0.8666 | 0.0791 | 0.8433 | 0.8245 | 0.8066 | 0.0916 | 0.7983 | 0.8217 | 0.7903 | 0.2072 | 0.6768 | 0.8979 | 0.8823 | 0.0773 | 0.8606 |
| Ours-V | 0.8862 | 0.8748 | 0.0644 | 0.8612 | 0.8263 | 0.8249 | 0.0696 | 0.8368 | 0.8752 | 0.8597 | 0.1139 | 0.8082 | 0.9177 | 0.8940 | 0.0416 | 0.8839 |

**[CQ3] Additional parameters (R2-Q4, R4-Q1):** Sorry for the unclear description. In fact, the baseline (with the ResNet50 backbone) is carefully designed to share a close number of parameters with the full model (178 MB vs 176 MB), which is adequate for proving the superiority of our CoADNet without introducing additional parameters.

**[R1-Q1] Video-SOD.** The input images in CoSOD task are not necessarily temporally-related, which deviates from Video-SOD that emphasizes temporal modelling. Hence, direct adaptation might not be applicable.

**[R1-Q2] Effectiveness of OIaSG.** Sorry for making the confusion. As mentioned in the ablation study, we have pre-trained the baseline for the SOD task on the DUTS dataset, which could prove the superiority of our OIaSG scheme.

**[R2-Q1] Unclear motivation.** 1) In the introduction, we have separately highlighted the three main motivations (please see Page 2, Line 41-62), which illustrate the necessity of the GASA, GGD, and GCPD modules item by item. We will make clearer statements for your concerns. 2) Our overall aggregation-and-distribution architecture for the problem of CoSOD is novel and brings very competitive performances. The GASA brings new insights in solving order-sensitivity and capturing long-range inter-image dependencies. Moreover, the GGD and GCPD further investigate group-individual interaction and co-saliency consistency that are very crucial but completely ignored in previous CoSOD methods.

**[R2-Q2] Missing related works (RW).** Due to limited space, we only analyzed the highly-related works [32,35,43,44]. Experiments included the most recent SOTA works for comparisons. We will add a RW section in the full version.

**[R2-Q3] Feature visualization.** The learned co-saliency features highlight the common and salient objects in each image, and suppress others. As visualized in Fig. 3, the features in the encoder show much higher response around co-salient objects with reduced background redundancy. We will further provide more visualizations for each module.

**[R2-Q5] Weak relevance.** This paper deals with CoSOD task under the NeurIPS track of *Applications -> CV*. Moreover, there have been some visual saliency researches on very recent NeurIPS's publications (e.g., [R1][R2]).

**[R3-Q2] Parameter selection & block-wise group shuffling.** 1) In practice, we tested several choices and found $B = 8$ works best. Actually, our model is not sensitive to $B$ within a reasonable range. We will discuss this parameter in the ablation study. 2) As depicted in Fig. 2, for the input $N$ images, we first split each feature map along channel axis into $B$ blocks, and concatenate all the $N$ blocks coming from the same $b^{th}$ partition.

**[R3-Q3] Order-insensitivity.** Order-sensitivity is caused by the sequential channel concatenation of individual features. In GASA, we apply channel-wise softmax to each shuffled features that are composed of several blocks, and then make element-wise summation of these blocks. In GCPD, we assemble the individual feature vectors and similarly apply softmax across channels and make summation. The two modified feature combination methods are order-invariant.

**[R4-Q2] Inconsistancy of [44] and VGG16 results:** The reported results in [44] adopts the VGG16 backbone. In our experiments, we tested the results of [44] provided by the authors, in which HRNet [R3] is used as backbone and hence causes inconsistency (our reported results are better). Although HRNet [R3] is stronger than VGG16 and ResNet50, our model (with VGG or ResNet backbone) still achieves superior performance. Please see **[CQ2]** for the VGG16 results .

**[R4-Q3] Idea of group semantics.** Our solution only shares a similar big picture with [32] in terms of aggregating group semantics. However, this paper explored new insights under a two-step aggregation-and-distribution framework. Instead of directly duplicating and concatenating the group semantics with individuals, we designed GGD for dynamic group-individual combination and suppression of distracting information redundancy, which turns to be very crucial but is ignored in previous studies. Besides, the GASA differs from [32] in attentive learning and long-range modelling.

**[R4-Q4] AP.** We list AP comparisons of [44] and ours in **[CQ2]**. We will report APs for all methods in the final version.

**[R4-Q5] Saliency priors.** In the CoSOD task, maintaining awareness of salient regions and knowing how to exploit saliency priors for co-saliency mining are critical. Compared with common practice of SOD pretraining, our OIaSG provides a more effective and flexible jointly-optimized workflow for integrating more reliable saliency guidance information, which is the first attempt for CoSOD. Ablation study also supports this.

**References:** [R1] M. Zhang, *et al.*, Memory-oriented decoder for light field salient object detection. *NeurIPS*, 2019.
[R2] T. Nguyen, *et al.*, DeepUSPS: Deep robust unsupervised saliency prediction via self-supervision. *NeurIPS*, 2019.
[R3] K. Sun, *et al.*, Deep high-resolution representation learning for human pose estimation. *CVPR*, 2019.

[Meta-Review · NeurIPS 2020]

This is a paper for which the reviewers failed to reach consensus, despite each reviewer having read and responded to author feedback in their review, and despite a discussion amongst the reviewers. Thus, I have read the paper myself and have also read each review carefully along with the author response, and will be including my own viewpoints of the points raised by the reviewers. The primary points raised in favor of the paper are well summarized by R3 in discussion, who wrote "However, what makes me feel good are the three instructive issues discussed by the authors and the corresponding solutions presented in this work. The proposed approach is not that hard to implement but can address the key problems in co-saliency detection and finally obtains the outperforming experimental results. So this work should be insightful to the co-saliency detection community." The primary point raised against the paper is a perception of lack of novelty, viewing the paper more as an integration of previous approaches. After careful thought and consideration of each reviewers thoughts on this topic, I have decided to down-weight this point for the following reasons. First, I do think it's important that every paper have a novel contribution -- but this does not necessarily mean a novel method. The contribution can be new results helping to understand previous methods more fully. Indeed, I personally feel that our field has far too much emphasis on "new methods" and far too little time or space given to verification and understanding of previous methods. So, I think that a paper that does a good job of selecting and integrating previous methods that work well together and showing the results on this can be seen as an important and useful contribution in this way. The second criticism raised is that there is some concern about the experimental setup. In discussion, R2 notes: Indeed, [44] also uses an extra SOD dataset (MSRA-B which contains 2,500 training images) to train their model. However, the proposed model relies on a much larger dataset i.e., DUTS, which includes 10,553 training images. Since the authors treat [44] as the baseline for comparison, I am not clear why they did not use the same train set. Previous works have show that using DUTS as train set can bring significant performance boost in salient object detection, so it is hard to know whether the performance improvements over [44] is from additional data or the proposed OIaSG." I believe the authors have attempted to address this issue in the supplementary material, which notes that "Note that, for fair comparison, the backbone feature extractor in baseline model is pretrained on a single-image saliency detection dataset (i.e., DUTS [3]) for shared feature learning." Furthermore, in their author response, they note "Effectiveness of OIaSG: Sorry for making the confusion. As mentioned in the ablation study, we have pre-trained the baseline for the SOD task on the DUTS dataset, which could prove the superiority of our OIaSG scheme." Thus, I think that the authors have indeed produced a reasonable apples-to-apples comparison environment giving the baseline method the same data to train on. However, I do expect the authors to clarify this explicitly in the final version of the paper so that it is easy to understand. As a minor point, I think that the discussion of related work that happens in the introduction would benefit from the signpost sub-heading "Related Work", to help readers navigate the paper. Finally, I am flagging to Senior Area Chair for additional review, given the unusually wide level of disagreement here.